# Brief communication: Significant biases in ERA5 output for McMurdo Dry Valleys region, Antarctica

Ricardo Garza-Girón[1,2*], Slawek M. Tulaczyk[1]

[1]Department of Earth and Planetary Sciences, University of California, Santa Cruz, CA, 95064, USA
[2]Department of Geosciences, Warner College of Natural Resources, Colorado State University, Fort Collins, CO 80523, USA.

*Correspondence to: Ricardo Garza-Girón (rgarzagi@ucsc.edu/r.garza_giron@colostate.edu)

**Abstract.** The ERA5 climate reanalysis dataset plays an important role in applications such as monitoring and modelling climate system changes in polar regions, so the calibration of the reanalysis to ground observations is of great relevance. Here, we compare the 2-metre air temperature time series of the ERA5 reanalysis and the near-surface bias-corrected reanalysis to the near-ground air temperature measured in 17 Automatic Weather Stations in the McMurdo Dry Valleys, Antarctica. We find that the reanalysis data has biases that change with the season of the year and that do not clearly correlate with elevation. Our results show that future work should rely on secondary observations to calibrate when using the ERA5 reanalysis in polar regions.

**Short Summary.** By analyzing temperature time series over more than 20 years, we have found a discrepancy between the 2-metre temperature values reported by the ERA5 reanalysis and the Automatic Weather Stations in the McMurdo Dry Valleys, Antarctica.

## 1 Introduction

ERA5 dataset represents the fifth iteration of ECMWF (European Center for Medium-Range Weather Forecasts) global climate hindcasting based on the Integrated Forecasting System (IFS) Cy41r2 derived by a combination of data assimilation and short-term simulations applying an operational numerical weather prediction (NWP) model (Hersbach et al, 2020). With its global coverage, high temporal resolution, and relatively high spatial resolution of 31 km, this dataset may prove particularly useful for research in polar regions such as Antarctica, where long-term climate observations are geographically sparse and often temporally discontinuous (Lazzara et al, 2012). A previous study found encouraging agreement between ERA5 output and AWS (Automatic Weather Station) data from 13 stations located in the southern section of Antarctic Peninsula (Tetzner et al., 2019). However, at least one other study has pointed out differences between ERA5 and selected weather stations across all Antarctica (Zhu et al., 2021).

Here, we report the results of a regional comparison between monthly 2-metre air temperatures in the McMurdo Dry Valleys region, Antarctica, reported in the ERA5 dataset and corresponding observations from 17 AWS locations. We focus our analysis on this region because of the relatively high spatial and temporal coverage of AWS observations and due to the high multidisciplinary research interest in this region which contains the main USA and New Zealand research stations and is proximal to Italian and Korean research stations.

Despite the encouraging results found by Tetzner et al. (2019) for the Southern Antarctic Peninsula, we find significant biases in the near-surface air temperatures measured at the AWS and the temperatures reported in the reanalysis datasets.

## 2. Data and methods

We analyze the daily surface temperature (2-metre temperature) recorded at 17 AWS (Figure 1) managed by the McMurdo Dry Valleys Long Term Ecological Research Project (LTER) since 1992, although some of the stations have been reporting data only since 1986 (Doran et al., 2002). Table 1 summarizes the AWS used in this study. We compare the AWS data to the monthly ECMWF ERA5 climate reanalysis surface temperature data (Muñoz Sabater, 2019) and we also tested against the near-surface bias-corrected reanalysis dataset (BCR) (Cucchi et al., 2022). The latter is obtained from applying the Water and Global Change (WATCH) forcing data methodology (Weedon et al., 2010) to the ERA5 dataset, which includes interpolating to a 0.5° × 0.5° grid and correcting for differences in elevation between the Climate Research Unit grid (New et al., 1999; 2000) and the ERA5 grid, along with other monthly-based biases corrections (Weedon et al., 2011, 2014; Cucchi et al., 2022). For each AWS, where daily 2-metre air temperature data was available, we ran a 30-day moving average filter with no overlap to obtain monthly time series. The ERA5 and BCR grid nodes used to compare to each individual AWS were selected by minimizing the distance between each AWS and all the nodes in the reanalysis grid (Figure 1). Finally, we interpolated both time series to a regular monthly sequence, and the time series for the ERA5 node data were truncated to match the periods where

data was available at their corresponding AWS. The elevation of the AWS and the nearest ERA5/BCR grid cells are often different, which can induce differences in the measured and calculated values of 2-metre air temperature. Therefore, we correct for the difference in altitude by applying a dry adiabatic lapse rate of 9.8 °C/km to the ERA5/BCR data, as done elsewhere (Bromwich et al., 2013). We report the mean temperature for the span of each time series and the standard error of the mean for each sample for the differences between the ERA5 and BCR datasets and the AWS with and without the altitude correction.

Furthermore, we compare the two data sets by analyzing the correlograms of the altitude-corrected temperatures and performing a linear regression. We report the squared correlation coefficients ($R^2$) as a metric of the goodness of fit and the p-values from the F-statistic to assess the level of statistical significance. Besides inspecting biases by making comparisons for all individual stations and their corresponding reanalysis grid cells, we also compare the overall mean temperature across all stations with the mean temperature across all grid cells within the main region of the McMurdo Dry Valleys (black box in Figure 1). We selected this region because it has the highest station density and including the stations outside of this box would imply using a much larger subgrid for the reanalysis that would not be truly representative of the area covered by the stations. This comparison is important given that the ERA5 and particularly the BCR grid cells might be too large to capture local phenomena such as topographic effects or seasonal temperature inversions at the AWS. Therefore, comparing average temperatures using the footprint of the whole region is different than calculating the average bias across all station, and it creates intuition on whether the individual elevation differences average out or not. We created median stacks of the temperature time-series for all AWS and for all the grid cells of the reanalysis that fall within the area. We interpolated all the data of the weather stations to a monthly time series, we patched with NaN (Not a Number) values the periods of time when data was not available (some stations have longer records than others) and we obtained a mean-stack of the time-series. Figure S1 shows the individual time-series for all AWS and all the ERA5 and BCR grid cells and their corresponding mean stack. We also tested using median stacks to analyze the effect of outliers in the data, but we did not find major differences between the mean and median stacks. Finally, we

use the difference between the median altitude of all weather stations and the median altitude of the selected
grid cells of each reanalysis product to apply the dry adiabatic lapse rate correction to the temperatures.
**3. Results**
Overall, the two reanalysis products show both cold and warm biases compared to the AWS temperatures.
Table 2 shows the results of the comparison at each station and the elevation map of the AWS as well as the
spatial distributions of the altitude-corrected biases are shown in Figure S2 and Figures S3 and S4,
respectively. We find that the biases in the ERA5 dataset are of smaller magnitude than the biases observed
for the BCR dataset. The altitude correction applied to the grid temperatures does not eliminate but reduces
the average bias across all stations. However, this is not the case for all stations; for ERA5, the altitude
correction increases the bias at three stations (FRSM, UHDM and VIAM), and for BCR the correction
increases the bias at five stations (BENM, BRHM, CAAM, FLMM and VIAM).
Contrary to the altitude-dependent biases found by Tetzner et al. (2019), our results do not show a clear
correlation between bias and elevation (see Figures S2, S3 and S4). Nevertheless, our results do suggest that
the ERA5 dataset has predominantly neutral to warm biases in the valleys, despite elevations, and neutral to
cold biases in the mountain ranges.
Figure 2 illustrates the comparison of the monthly temperature time series for one of 17 locations used in this
study (Lake Vida) and the temperatures from the ERA5 and BCR datasets over the time span of more than
two decades. In this case, the monthly temperature mismatch between the AWS and the ERA5 and BCR
altitude-corrected temperatures is particularly large during the winter months, when observations indicate
actual temperatures were about 10°C lower than ERA5 or BCR temperatures (Figure 2c,d). All the
correlograms shown in Figures S5-S21 suggest that there is a strong seasonality in the relationship between
the data sets. During the austral winter and summer seasons the temperatures are generally closely clustered
together, systematically being more correlated during the winter and more dispersed during the summer. The
spring and fall seasons show a hysteresis that is repeated over all the comparisons. As the environment warms
up during the spring months the ERA5 and BCR temperatures are above the best-fit line and drop below it
during the fall. These seasonal biases may ultimately be helpful in revealing what climate processes must be
better represented in the ERA5 reanalysis to eliminate the observed temperature biases.
The comparison between the average stack of the AWS with the ERA5 and BCR temperatures for the selected
subregion (black box in Figure 1) is shown in Figure 3. Interestingly, our regional average analysis suggests
that the altitude corrected ERA5 temperatures (black line in Figure 3) have a cold bias of -9.6 ± 1.0 °C
compared the AWS temperatures, but the altitude corrected BCR temperatures (black dashed line in Figure
3) are much closer to the AWS temperatures. Nevertheless, the BCR temperatures do show a warm bias of
3.3 ± 1.0 °C.
**4. Discussion**
Our results differ significantly from the findings reported by Tetzner et al. (2019) for the Southern Antarctic
Peninsula - Ellsworth Land region. For that region there is a slight cold bias of the ERA5 surface temperatures
close to the coast (-0.51 ± 0.74 °C) and a slight warm bias in the mountain range escarpment (0.14 ± 0.72
°C) which has encouraging implications for using the reanalysis data where there is no AWS coverage, which
represents most of Antarctica. In contrast, we find no obvious topographic dependence on the temperature
differences between AWS and ERA5 data. Averaged over the whole region, the altitude-corrected
temperatures of the ERA5 dataset have a slight cold bias of -0.4 ± 0.8 °C, whereas the BCR data has a cold
bias of larger magnitude (-4.4 ± 1.9 °C). However, there are large variations from one site to another, and
from one season to another. Some of the large cold biases for the altitude corrected ERA5 and BCR data are
observed during the summer months, with average differences up to -4.9 ± 0.1 °C and -16.2 ± 0.3 °C,
respectively. This may be a particularly significant problem given the fact that warm summer temperatures
determine the annual melt rate of snow, glaciers, and permafrost in Antarctica. Modelling of snow or ice
melting driven by ERA5 temperatures (e.g., Costi et al., 2018) with a strong cold bias, as observed in our
study region, will result in a significant underestimate of summer melt production. Conversely, many stations

show a warm bias during the winter months, which could potentially be related to temperature inversions that create air parcels with negative buoyancy and drive katabatic winds down the glacial streams and valleys (Phillpot & Zillman, 1970).

The differences in the regional averaged temperature time series for the AWS and the ERA5 and BCR renalyses do show different biases than the ones reported above, which are based on the average difference between each AWS and their corresponding grid cell. For the average stacks, the ERA5 temperatures are significantly colder than the mean AWS temperatures and the BCR temperatures are slightly warmer, and they have an overshoot during the Summer and the Winter alike. This finding is interesting and suggests that the BCR reanalysis might be a better reference for the Dry Valleys region when studying a large area, but that the ERA5 reanalysis might be a better model for more local targets.

In general, our findings agree with the findings of Zhu et al. (2021) in that they also find a cold bias for West Antarctica. However, our results highlight the degree in which such biases can be found at a regional and local scale and by using different datasets. As in situ instrumentation increases in the future in the McMurdo Dry Valleys, future research on the topic could illustrate in more detail the sources of the biases between reanalyses products and weather stations reported here. Particular attention should be given to the effect on topography and seasonal temperature inversions at smaller scales. Although the ERA5 reanalysis and its bias-corrected version are outstanding sources of global climate variables, the discrepancy between our results and those obtained by Tetzner et al. (2019) suggests that secondary observations should be used to test the reliability of the ERA5 and BCR dataset in polar regions, particularly when performing studies at scales shorter than 0.5°.

## 5. Conclusions

We have compared the surface temperature (2-metre temperature) recorded at 17 AWS in the McMurdo Dry Valleys, Antarctica with temperatures from the ERA5 reanalysis dataset. We found that the temperatures reported by the global climate reanalysis and its bias-corrected version can have significant warm and cold

biases relative to the weather stations. The cold temperature bias appears to be the largest during the warm summer months, when loss of snow and ice to melting is the largest. Warm biases are more common during the winter months, when atmospheric temperature inversions are common. When using the average temperature across many stations in a region and compared to the average temperature of all the grid cells in that region, the bias corrected reanalysis shows a slight warm bias, whereas the ERA5 temperatures show a significant cold bias. We advise using secondary observations to assess the accuracy of parameters included in ERA5 reanalysis and its bias corrected version for polar regions when performing studies at different scales.

*Data availability*. The AWS data were provided by the NSF-supported McMurdo Dry Valleys Long Term Ecological Research program (OPP-1637708) and can be accessed at: https://mcm.lternet.edu/meteorological-stations-location-map. The "ERA5-Land hourly data from 1950 to present" (DOI: 10.24381/cds.e2161bac) and the "Near surface meteorological variables from 1979 to 2019 derived from bias-corrected reanalysis" (DOI: 10.24381/cds.20d54e34) were downloaded from the Copernicus Climate Change Service (C3S) Climate Data Store.

*Author contributions*. ST conceived the study. RGG performed the analysis. RGG and ST prepared the manuscript with equal contributions.

*Competing interests*. The authors declare that they have no conflict of interest.

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

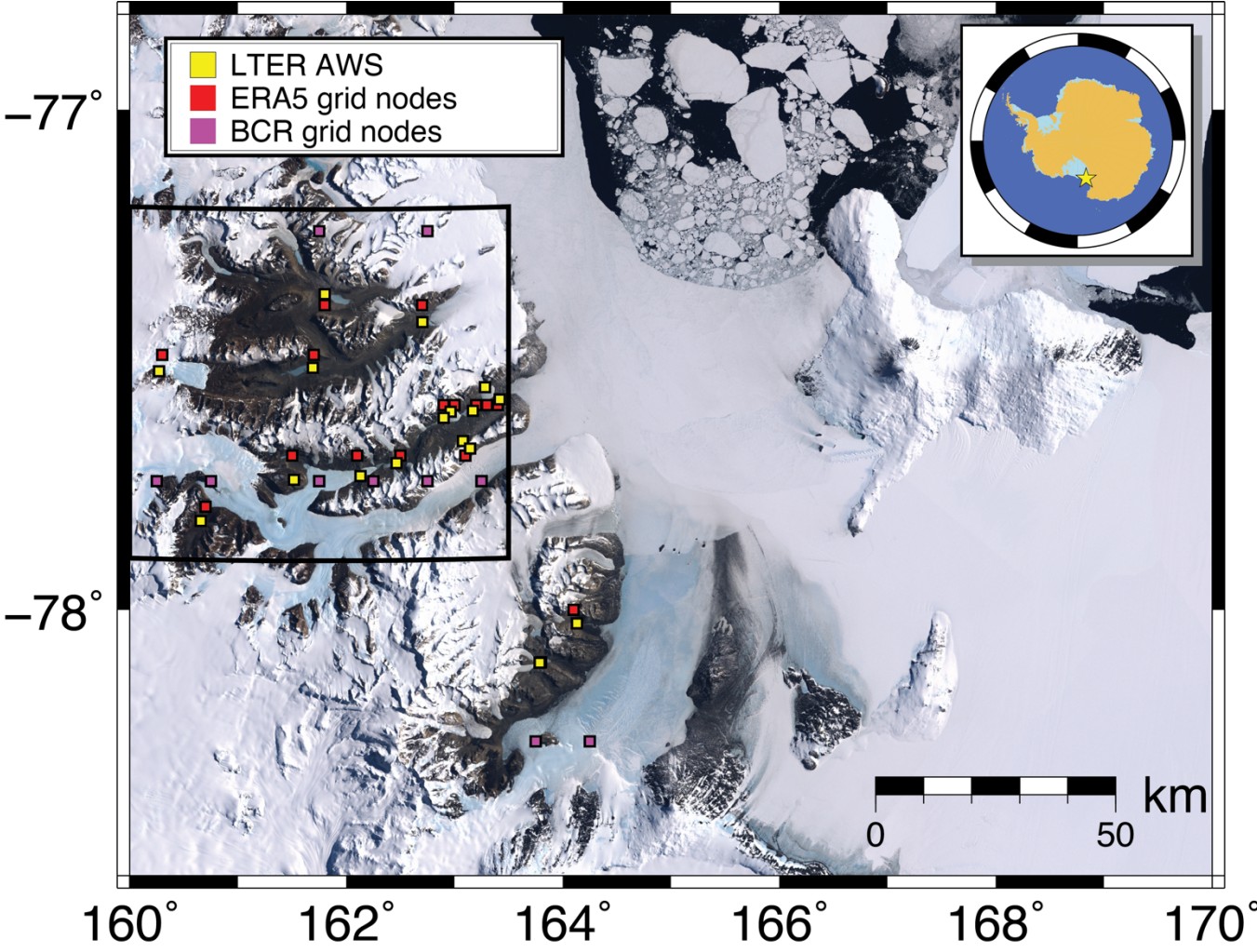

**Figure 1. Map of the McMurdo Dry Valleys region**. The location of the AWS managed by LTER is shown with yellow squares and their corresponding closest ERA5 and BCR grid nodes are shown with red squares and magenta squares, respectively. The black box represents the area where regional averages for all AWS and ERA5 and BCR grid cells were calculated. The distance to the sea and the topography of the region can be appreciated in the background satellite image.

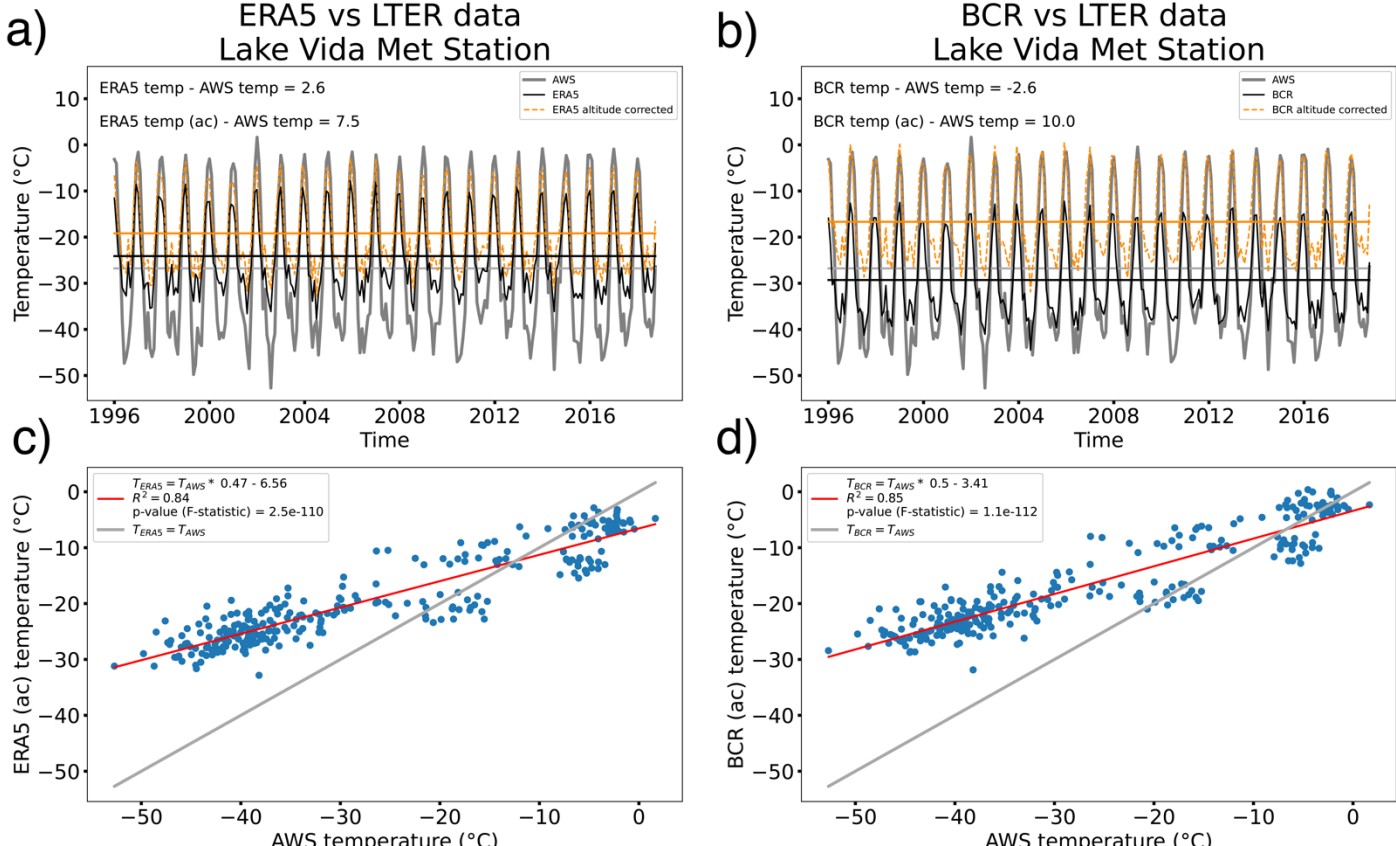

**Figure. 2 Comparison of the monthly averaged 2-metre air temperatures recorded at station Lake Vida (VIAM) and the values from the closest grid node of the ERA5 and BCR datasets.** Time series of the AWS data (grey curve) compared to the reanalysis data (black curve) and the altitude-corrected (ac) reanalysis data (dashed orange curve) for the ERA5 (a) and BCR (b) datasets. The correlograms showing the best fit line (red line) to the relationship between the AWS temperatures and the ERA5 and BCR temperatures are shown in (c) and (d), respectively. Note the seasonal variation in the relationship, particularly the large bias during the winter months.

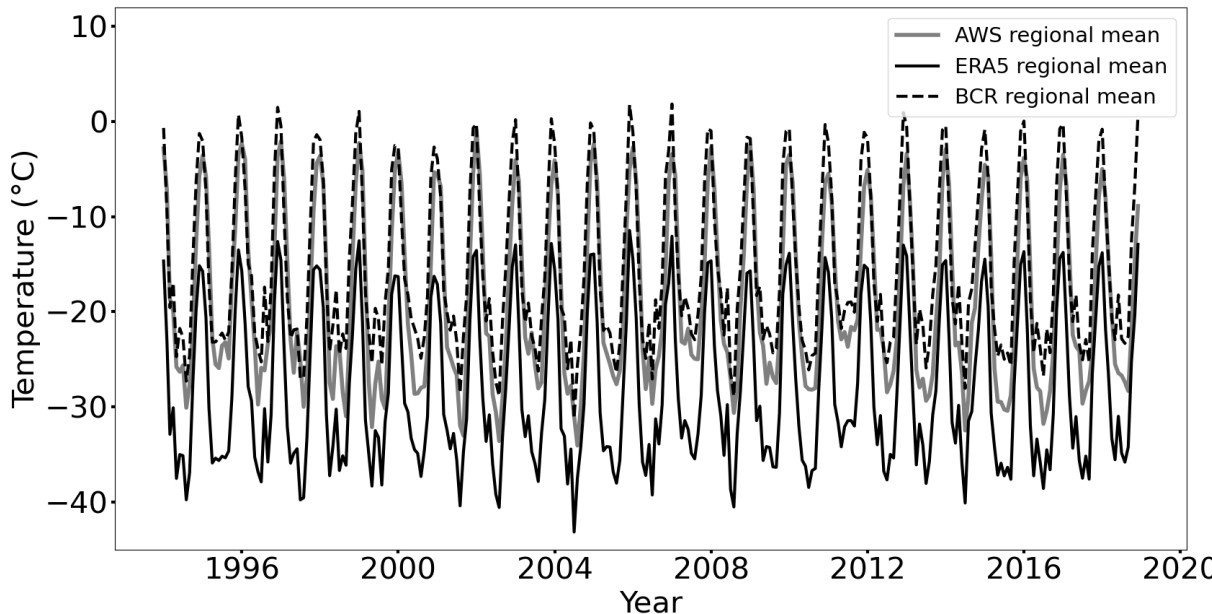

**Figure. 3 Regional mean stacks comparison for a subarea of the McMurdo Dry Valleys (black box in Figure 1).** The average time series of temperatures across all stations is shown with a thick gray line, the average temperature from the ERA5 grid cells that are within the region is shown with a black line, and the average temperature from the BCR subgrid is shown with a black dashed line.

| AWS Location name | AWS ID | Latitude | Longitude | Elevation (m.a.s.l.) |
|---|---|---|---|---|
| Beacon Valley | BENM | -77.828 | 160.6569 | 1,176 |
| Lake Bonney | BOYM | -77.7147 | 162.4646 | 64 |
| Lake Brownworth | BRHM | -77.4344 | 162.7036 | 279 |
| Canada Glacier | CAAM | -77.6133 | 162.9644 | 264 |
| Commonwealth Glacier | COHM | -77.5646 | 163.2823 | 290 |
| Explorer's Cove | EXEM | -77.5887 | 163.4175 | 25 |
| Mt. Fleming | FLMM | -77.5327 | 160.2714 | 1,870 |
| Lake Fryxell | FRLM | -77.6113 | 163.1701 | 19 |
| Friis Hills | FRSM | -77.7474 | 161.5162 | 1,591 |
| Garwood Ice Cliff | GAFM | -78.0259 | 164.1315 | 51 |
| Howard Glacier | HODM | -77.6712 | 163.0773 | 472 |
| Lake Hoare | HOEM | -77.6254 | 162.9005 | 77 |
| Miers Valley | MISM | -78.1011 | 163.7877 | 51 |
| Taylor Glacier | TARM | -77.74 | 162.1314 | 334 |
| Upper Howard | UHDM | -77.686 | 163.145 | 826 |
| Lake Vanda | VAAM | -77.5257 | 161.6913 | 296 |
| Lake Vida | VIAM | -77.3778 | 161.8007 | 351 |

**Table 1.** List of available AWS in the McMurdo Dry Valleys region.











| AWS Location name | AWS ID | Distance to closest ERA5 node (km) | AWS data date range | Average 2 m air temperature @ AWS | Average 2 m air temperature @ ERA5 node / altitude corrected | Average 2 m air temperature @ BCR node /altitude corrected | $ERA5_{mean\_temp}$ - $AWS_{mean\_temp}$ / $ERA5 (ac)_{mean\_temp}$ - $AWS_{mean\_temp}$ | $BCR_{mean\_temp}$ - $AWS_{mean\_temp}$ / $BCR (ac)_{mean\_temp}$ - $AWS_{mean\_temp}$ |
|---|---|---|---|---|---|---|---|---|
| Beacon Valley | BENM | 3.27 | 2000-12-11 - 2012-11-19 | -21.5 ± 0.7 | -33.5/-24.2 ± 0.7 | -29.4/-38.3 ± 0.7 | -12.1/-2.8 ± 1.4 | -8.0/-16.8 ± 1.4 |
| Lake Bonney | BOYM | 1.84 | 1993-12-08 - 2018-10-09 | -17.2 ± 0.6 | -24.0/-13.3 ± 0.4 | -29.3/-20.7 ± 0.5 | -6.7/3.9 ± 1.0 | -12.1/-3.4 ± 1.1 |
| Lake Brownworth | BRHM | 3.83 | 1995-01-23 - 2018-11-10 | -19.9 ± 0.7 | -25.4/-20.0 ± 0.5 | -29.3/-31.0 ± 0.5 | -5.5/-0.1 ± 1.2 | -9.4/-11.1 ± 1.2 |
| Canada Glacier | CAAM | 1.71 | 1994-12-18 - 2011-01-05 | -16.3 ± 0.7 | -23.1/-18.8 ± 0.6 | -29.3/-30.9 ± 0.6 | -6.7/-2.5 ± 1.3 | -13.0/-14.5 ± 1.3 |
| Commonwealth Glacier | COHM | 3.96 | 1993-12-06 - 2018-10-30 | -17.6 ± 0.5 | -22.1/-21.1 ± 0.5 | -29.3/-16.1 ± 0.5 | -4.4/-3.4 ± 1.0 | -11.6/-1.6 ± 1.0 |
| Explorer's Cove | EXEM | 1.32 | 1997-12-05 - 2018-11-23 | -18.9 ± 0.7 | -21.7/-19.0 ± 0.5 | -9.3/-13.5 ± 0.5 | -2.7/0.0 ± 1.2 | -10.3/5.5 ± 1.2 |
| Mt. Fleming | FLMM | 3.7 | 2011-01-22 - 2018-11-11 | -24.2 ± 0.6 | -34.0/-23.5 ± 0.8 | -29.2/-35.9 ± 0.8 | -9.8/-0.7 ± 1.4 | -5.0/-11.7 ± 1.4 |
| Lake Fryxell | FRLM | 1.45 | 1994-12-12 - 2018-11-19 | -19.7 ± 0.7 | -22.4/-17.8 ± 0.5 | -29.3/-13.4 ± 0.5 | -2.6/2.0 ± 1.2 | -9.5/6.4 ± 1.2 |
| Friis Hills | FRSM | 5.28 | 2011-01-04 - 2018-11-06 | -22.5 ± 0.6 | -26.8/-28.6 ± 0.7 | -29.2/-28.7 ± 0.8 | -4.3/-6.0 ± 1.3 | -6.6/-6.2 ± 1.4 |
| Garwood Ice Cliff | GAFM | 2.97 | 2012-01-24 - 2012-12-19 | -16.6 ± 2.8 | -23.6/-17.7± 2.3 | -30.7/-29.6 ± 2.3 | -7.0/-1.0 ± 5.1 | -14.0/-12.9 ± 5.1 |
| Howard Glacier | HODM | 3.25 | 1993-12-04- 2018-10-31 | -17.18 ± 0.4 | -20.8/-20.3 ± 0.5 | -29.3/-17.9 ± 0.5 | -3.6/-3.1 ± 0.9 | -12.1/-0.7 ± 0.9 |
| Lake Hoare | HOEM | 2.82 | 1987-11-25 - 2018-11-29 | -17.61 ± 0.5 | -23.5/-15.9 ± 0.4 | -29.2/-28.9 ± 0.4 | -5.9/1.7 ± 0.9 | -11.6/-11.3 ± 0.9 |
| Miers Valley | MISM | 0.31 | 2012-02-11 - 2018-11-06 | -16.69 ± 1.00 | -23.2/-18.2 ± 0.9 | -29.5/-20.0 ± 0.9 | -6.6/-1.5 ± 1.9 | -12.8/-3.3 ± 1.9 |
| Taylor Glacier | TARM | 4.51 | 1994-12-05 - 2018-11-05 | -16.9 ± 0.5 | -25.4/-15.1 ± 0.4 | -29.3/-23.3 ± 0.5 | -8.5/1.8 ± 0.9 | -12.4/-6.4 ± 1.0 |
| Upper Howard | UHDM | 1.89 | 2001-11-28 - 2003-12-24 | -16.56 ± 1.5 | -20.3/-23.3 ± 1.7 | -28.7/-20.8 ± 1.7 | -3.7/-6.8 ± 3.2 | -12.2/-4.2 ± 3.2 |
| Lake Vanda | VAAM | 2.87 | 1994-12-08 - 2018-12-07 | -19.58 ± 0.7 | -25.1/-17.4 ± 0.4 | -29.2/-16.1 ± 0.5 | -5.5/-2.2 ± 1.2 | -9.6/3.5 ± 1.1 |
| Lake Vida | VIAM | 2.47 | 1995-12-08 - 2018-11-14 | -26.68 ± 1.0 | -24.1/-19.2 ± 0.5 | -29.3/-16.7 ± 0.5 | 2.6/7.5 ± 1.5 | -2.6/10.0 ± 1.5 |

**Table 2.** List of comparison results between the temperatures recorded at the AWS and the closest ERA5
and BCR nodes. For each of the reanalysis datasets, we show the reported 2 m air temperature and the
altitude-corrected (ac) value and their comparison to the average temperature at the AWS.