# Peer review of "Brief communication: Significant cold bias in ERA5 output for"

_The Cryosphere, 2023_

## Author Comment (AC2)

We would like to thank the editor for the handling of our manuscript. We would also like to thank the reviewers for their comments and suggestions, which, without a doubt, have helped with the improvement of the manuscript by clarifying important points such as the difference between the altitude of the weather stations and the reanalysis grid.

We respond to your comments in more detail below:

**Reviewer 1**

Comments:

Lines 41 – 43.  Are the datasets independent, so that the observations have not been assimilated into the ERA5 reanalysis? Could this be explicitly mentioned in the text?

R: The ERA5 dataset and the bias corrected reanalysis dataset are not completely independent since the latter is obtained from applying an elevation (and other monthly-based climate biases) corrections. We now specify this in the text.

Lines 41 – 43.  How large is the area where these 17 AWSs are located?

R: There was an error upon submission of the manuscript and Figure 1, which shows the map of the location of the stations and the scale of the area, was not included. We apologize for this. The manuscript has now the correct Figure 1.

Lines 47 – 49. Are all observations compared with the same grid cell in ERA5?

R: Each AWS was compared to the closest ERA5 grid node. The wording in the manuscript was wrong and did give the impression that we only used one node. We have fixed the text to clarify this, it now reads:

"The ERA5 grid nodes used in comparisons to each individual AWS were selected by minimizing the haversine distance between each AWS and all the nodes in the reanalysis grid."

Lines 59 – 62. How representative are observation sites and are geographical properties similar in ERA5 as in observation sites? I think you could write a sentence or two about how similar surface properties are around the stations and in ERA5 because surface geographical properties remarkably affect near surface temperature in the polar regions. As surface inversions are common, near surface temperatures are sensitive to even small scale features in surface topography.

R: We hope that the corrected Figure 1 can bring some insight into answering this question. The AWS geographical locations range from a variety of distances from the sea and they also vary in elevation from almost at sea level to about 1.8 km altitude. In this sense, they are geographically representative of the Dry Valleys region. As can be seen in Figure 1, the closest ERA5 grid nodes are quite well correlated in space with the AWS. However, this is not the case with the BCR dataset. This is due to the fact that the ERA5 grid has a resolution of 0.1° X 0.1°, whereas the BCR was downsampled to a grid resolution of 0.5° X 0.5°.

Lines 59 – 62. What is the altitude range in ERA5 grid cells? Is the surface elevation almost the same in reanalysis as in reality in an observation site?

R: This is a very good question. We were previously, erroneously, missing applying a lapse-rate correction for the difference in altitude between the AWS and the closest grid cell. As you can see in the modify version of the manuscript, we now apply this correction and in general it reduces the bias but it does not completely eliminate it. Also, interestingly, there are a few cases where the altitude correction makes the bias even worse. Given all of this, we now present the results of the two datasets (ERA5 and BCR) with and without the altitude corrections.

Discussion. Could you suggest any reason for negative bias in ERA5? If you have any ideas why temperatures are underestimated in ERA5, you could maybe a little bit speculate with them in discussion. Especially, why ERA5 perform well in the Southern Antarctic Peninsula - Ellsworth Land region in the study of Tetzner et al. (2019) but not in the McMurdo area?

R: There are a number of possibilities of why there could be a negative bias in the McMurdo Sound compared to the South Antarctic Peninsula-Ellsworth Land. For example, the Peninsula is further north than the McMurdo Sound; there is a much greater mass of sea ice that accumulates in the McMurdo Sound compared to the Peninsula (which we suspect is an important effect in the bias); the difference could also be due to differences in regional climatic conditions, such as cloud coverage.

However, since these are all speculative explanations, we will refrain from adding them to the text.

Lines 89 – 90. "TheERA5 temperatures show a large overshoot during the summer, with an average difference of 6.7 ± 0.8 °C (e.g., Figure 2)." What do you mean with overshoot? A large negative bias?

R: Yes, here we mean that during the summer months the bias is larger than compared to the rest of the year. We have removed the word "overshoot" and rephrased the paragraph to avoid confusion.

Figure 1. Figure is wrong. According to text and caption the figure should show a map of the region but there is a time series of temperature in the figure. I think the map could give answers to some of my questions. Therefore, I hope that the map includes information about surface conditions (i.e. satellite image in summer at background) topography (i.e. surface elevation contours), sizes of ERA5 grid cell (grid cell boundary lines) and locations of AWSs.

R: We apologize, we have added the correct Figure 1. Indeed, figure 1 was always meant to show not only the location of the stations and the ERA5 nodes, but also the satellite image of the McMurdo sound where the topography of the Dry Valleys region can be appreciated.

Figure 2b. Could you use the same temperature scale in both axes or add line T_obs = T_ERA to help comparison between ERA5 and observations.

R: Thank you for the suggestion, we have added a T_obs = T_era line in the correlograms of figure 2 (and in all subsequent supplementary figures). It does help show the biases more clearly.

**Reviewer 2**

The manuscript presents evaluation of ERA5 reanalysis 2-m temperatures (T2m) in Antarctica. The topic is important, as ERA5 is very much applied in climate research, and is often considered to represent the best available information from regions with no or sparse observations avaialbale. However, the manuscript suffers from major shortcomings, and requires a major revision before eventual acceptance.

Major comments

1. The difference between observations and ERA5 T2m is surprisongly large and the bias is negative (cold), although previous studies from Polar regions have indicated predominantly positive (warm) biases. This makes me to suspect that the difference between the real ice sheet elevation and that in ERA5 has not been taken into account.

This is a fundamental shortcoming, as it does not make sense to compare observed and reanalysis-based T2m values if the model orography does not fit with the true orography. See, e.g., papers by Bromvich group on how to take the elevation difference into account comparison of observations and model products. First, the authors should quantify the elevation difference between each observation site and the nearest reanalysis grid cell. Note that even if the ERA5 biasdoes not depend on the topography (lines 85-86), it may well depend on the elevation difference between ERA5 grid and reality.

R: Thank you very much for your comment. Indeed, we were erroneously missing applying the lapse-rate correction for the differences in altitude from the AWS and the ERA5 grid. We have applied the correction using the standardly used value of 6.5 °C/km. The altitude correction does reduce the average bias observed in the region, but it does not completely eliminate it.

2. It seems that the authors do not fully understand the concept of an atmospheric reanalysis. A reanalysis is not derived by a combination of climate data assimilation and climate simulations (lines 24-25) but from a combination of data assimilation and short-term (mostly 6 hours) simulations applying an operational numerical weather prodiction (NWP) model. Also the text on lines 77-79 may give an impression that ERA5 is applied in seasonal-scale simulations, which is not the case, and the summertime ERA5 T2m values do not have any effect on the annual melt rate of snow, glaciers, and permafrost in Antarctica (i.e., text on lines 91-92 is misleading). The authors are correct that modelling of snow and ice melting using ERA5 temperature as atmospheric forcing is a problem, if ERA5 T2m has a large bias.

R: We have rephrased the definition of the ERA5 reanalysis, the text now reads:

"ERA5 dataset represents the fifth iteration of ECMWF (European Center for Medium-Range Weather Forecasts) global hindcasting based on the Integrated Forecasting System (IFS) Cy41r2 derived by a combination of data assimilation and short-term simulations applying an operational numerical weather prediction (NWP) model (Hersbach et al, 2020). "

Regarding the seasonal variation, it seems that the wording in our text was confusing, since we do not suggest that the ERA5 is applying in seasonal-scale simulations, but rather that when analyzing the decades-long time series and the correlogram, it is clear that the is a

seasonal dependency on the bias. The correlogram shows a hysteresis that is present for all stations, and the larger bias during the summer season is also evident from the time series and the correlograms. We have rephrased our text with the hope that this is now more clear.

3. Figure 1 is missing, and Figure 2 is presented twice.

R: We apologize, there seems to have been a mistake upon submission. We have now added the correct Figure 1.

Minor comments

4. Specify what you mean by "ground temperatures" (line 32)

R: By "ground temperatures" we referred to the 2-m air temperatures, so it was misleading and we have removed it from the text.

5. Better explain the near-surface bias corrected reanalysis dataset (line 45)

R: We added an explanation of the near-surface bias corrected reanalysis dataset and added a number of references where the dataset and the corrections applied are documented in detail.

6. In the rightmost column of Table 1, give the difference reanalysis minus observations so that positive biases are seen as positive values. Consider the observation accuracy of automatic weather stations in Antatrctica, and remove the irrelevant second decimales from the temperature data.

R: We have changed the order of the comparisons to show the reanalaysis minus the AWS data. We removed the second decimals and we also now include the altitude corrected values for both ERA5 and the bias-corrected reanalaysis datasets to show the effect of the correction.

---

## Referee Report (RR1)

Authors have satisfyingly responded most of my questions and concerns and the manuscript is almost on the sufficient level for publishing. However, I am totally satisfied the methods how different elevations of model grid cell and observational site are taken into account. Typically in the atmosphere temperature decreases upward and the lapse rate correction 6.5 °C/km perhaps mostly leads a reasonable correction. However, in the polar region especially in winter, temperature inversions are common and therefore temperature often increases upwards. Occurrence of inversions also amplify the effect of local topography on near surface temperature as the coldest airmass pour in the valleys and near surface temperature are often remarkably higher on slopes and tops of hills or mountains than on valleys.

Overall, it is challenging to compare model products directly with observation because they represent different things. Model product represents average over the whole grid cell and observation might be representative only near observational site. Complex surface topography and frequently occurring temperature inversion makes direct comparison between observations and model product even more difficult.

My suggestion is at least add some discussion about effects of stratification on elevation correction or calculate correction coefficient utilizing specific lapse rate for seasons. You may use observed temperatures to estimate specific lapse rate correction for the area and each season as the observational site are located in different elevation but horizontally relatively close to each other. However, small scale surface topography can still cause large differences between observed and modelled temperatures.

Overall, in my opinion, the manuscript can be published after adding thorough thinking of the effects of stratification and local surface topography on differences between observations and model fields in the manuscript.

---

## Author Response (AR2)

We thank you for your thoughtful reviews and responses to our revised version of the manuscript. Both reviewers raised relevant points regarding the altitude correction to the temperature differences and provided great suggestions on what could be causing the observed biases.

We respond to your comments in more detail below:

**Reviewer 1**

The authors have reasonably well addressed most of my comments. However, my most important comment on the altitude correction is not properly addressed. The authors have done the correction using a moist-adiabatic lapse rate of 6.5 K/km, which is a typical value for the troposphere as a global average. It differs from the dry-adiabatic lapse rate due to the release of latent heat of condensation, which often occurs when air parcels rise from the Earth surface. However, the Antarctic atmosphere is very dry, not least in the region of the McMurdo Dry Valleys addressed in the manuscript. Hence, there is typically no condensation and, even if there sometimes is, the amount of water vapour condensed is so small that it seldom has detectable effects on the lapse rate. Accordingly, the dry-adiabatic lapse rate of 9.8 K/km has to be applied in the altitude correction, as done, e.g., in Bromwich et al. (2013). This will have a large effect on the results and conclusions of the manuscript.

Thank you for pointing out this relevant issue. Indeed, the use of a dry adiabatic lapse rate correction is more appropriate for the Antarctic continent and particularly for the Dry Valleys region which is known for its cold, arid climate. By running our analysis utilizing the lapse rate correction of 9.8 K/km the regional averaged bias is low for the ERA5 dataset and still significant for the bias-corrected product. However, as the new version of the manuscripts highlights, the biases observed at independent stations suggest that there can be significant differences in the overall temperatures of the AWS and the reanalysis products and that these differences have a seasonal dependence. Our results also suggest that topography is not a deterministic factor on the bias. Ultimately, our goal is to report these discrepancies with the hope that it will create awareness on the direct use of ERA5 data for certain applications and perhaps open a border discussion on the phenomenology that causes the seasonal variations.

Reference: Bromwich, D. H., F. O. Otieno, K. M. Hines, K. W. Manning, and E. Shilo (2013), Comprehensive evaluation of polar weather research and forecasting model performance in the Antarctic, J. Geophys. Res. Atmos.,118, 274–292,doi:10.1029/2012JD018139.

Thank you for the suggestion, we have added the citation to the manuscript.

**Reviewer 2**

Authors have satisfyingly responded most of my questions and concerns and the manuscript is almost on the sufficient level for publishing. However, I am totally satisfied the methods how different elevations of model grid cell and observational site are taken into account. Typically in the atmosphere temperature decreases upward and the lapse rate correction 6.5 °C/km perhaps mostly leads a reasonable correction. However, in the polar region especially in winter, temperature inversions are common and therefore temperature often increases upwards. Occurrence of inversions also amplify the effect of local topography on near surface temperature as the coldest airmass pour in the valleys and near surface temperature are often remarkably higher on slopes and tops of hills or mountains than on valleys.
Overall, it is challenging to compare model products directly with observation because they represent different things. Model product represents average over the whole grid cell and observation might be representative only near observational site. Complex surface topography and frequently occurring temperature inversion makes direct comparison between observations and model product even more difficult.
My suggestion is at least add some discussion about effects of stratification on elevation correction or calculate correction coefficient utilizing specific lapse rate for seasons. You may use observed temperatures to estimate specific lapse rate correction for the area and each season as the observational site are located in different elevation but horizontally relatively close to each other. However, small scale surface topography can still cause large differences between observed and modelled temperatures.
Overall, in my opinion, the manuscript can be published after adding thorough thinking of the effects of stratification and local surface topography on differences between observations and model fields in the manuscript.

Thank you for your comments. We agree that it is challenging to have a direct comparison of the temperatures from the climate reanalysis products and the temperatures measured at the AWS. This is precisely the objective of our manuscript, to emphasize the existence of biases at a local scale and that these differences have a seasonal dependence, so that further studies proceed with caution when using the reanalysis datasets.

By using the dry adiabatic lapse-rate correction suggested by Reviewer 1, which is indeed appropriate for Antarctica, the seasonal biases became quite evident, even when the

regional average decreased. We added to the discussion arguing that the warm winter biases can be due to temperature inversions (thank you for pointing this out). However, we refrained from calculating season-dependent bias corrections, since the scope of our manuscript is to show the existence of such biases and suggesting that future work should proceed with caution when using ERA5 data as a direct metric of near-surface temperatures.

---

## Author Response (AR3)

We thank you once again for your reviews and responses to our revised version of the manuscript.

We respond to your comments below:

**Reviewer 1**

The manuscript has clearly improved from the previous version. Effects of inversion on differences between 2m-temperature between observations and reanalyses has briefly been discussed but it is still unclear how large part of differences between observation and ERA5 or BCR reanalysis is associated with real biases in reanalysis and how large part of biases is associated with effects local conditions (subgrid scale variation of temperature) which cannot be captured by reanalyses.

On one hand, the temperatures in reanalyses represent the mean temperature in grid box but there could be a large variation in 2m-temperature inside each grid box which cannot be captured by reanalyses, and location of observation stations are probably not always representative for the whole grid cell. On the other hand, the correlograms in the supplement and the sentence in lines 78 – 80 " Nevertheless, our results do suggest that the ERA5 dataset has predominantly neutral to warm biases in the valleys, despite elevations, and neutral to cold biases in the mountain ranges " suggest that ERA5 probably have on average too weak inversion in winter which may explain the differences between observation and ERA5 reanalysis and cause real bias in 2m-temperatures in ERA5. The complex topography and inversion amplify local variations in near surface temperature. Therefore I think, It would be clarifying to see the mean seasonal difference between analysis and over all observation stations. Even though a single station is not representative for average condition of the grid box, a relatively large sample of stations in this relatively small area could probably well present the mean conditions in the area. Therefore, I think that a figure which shows the seasonal cycle of mean difference between all stations and reanalyses would useful to understand biases in the reanalyses.

Overall, I could recommend publishing the manuscript after minor revision.

Thank you for the thoughtful suggestion. Indeed, including the figure showing the mean difference between all stations and the ERA5 and BCR reanalyses gives a different perspective to understand the biases. We addressed this by stacking the interpolated time series of all stations and all the grid cells within a box in the Dry Valleys (now shown in Figure 1). We are also adding a figure to the supplement to show the individual stacks, which are generally coherent.

Interestingly, this new analysis shows that the BCR version has a small warm bias and the ERA5 reanalysis has a strong cold bias when considering all the region. Even though this new piece of analysis does not elucidate the reasons behind the biases, it does add to the scope of our manuscript, which is to simply report the existence of these biases in the McMurdo Dry Valleys at different scales so that other researchers can consider them in the future. Certainly, future research, hopefully with the help of an increasing number of AWS, might help determine the exact causes of these biases at different locations.

**Reviewer 2**

The revised version of the manuscript is better but still includes some issues that should be clarified. On lines 49-51 it is stated that BCR is obtained applying the WATCH forcing data methodology to the ERA5 dataset, which includes an elevation correction. However, it remains unclear if this elevation correction differs from the dry-adiabatic one applied to ERA5 in this study. To interpret the reasons for the differences between the AWS data set and the BCR and ERA5 products, and the possible dependence of the differences on elevation, it is really important to clarify if the elevations corrections applied are similar or different. The Discussion section has to be extended accordingly.

Thank you for pointing this out. We have clarified that the elevation correction applied to the bias-corrected reanalysis is done based on the difference between the grid used by the Climate Research Unit (CRU) and the ERA5 grid. We also added two citations for referencing the CRU grid and further details on the corrections applied to the bias-corrected version of the ERA5 reanalysis can be found in Cucchi et al., 2022.

Cucchi M., Weedon G. P., Amici A., Bellouin N., Lange S., Müller Schmied H., Hersbach H., Cagnazzo, C. and Buontempo C.: Near surface meteorological variables from 1979 to 2019 derived from bias-corrected reanalysis, version 2.1, Copernicus Climate Change Service (C3S) Climate Data Store (CDS), 10.24381/cds.20d54e34, 2022.

New, M., Hulme, M., and Jones, P.: Representing Twentieth-Century Space–Time Climate Variability. Part I: Development of a 1961–90 Mean Monthly Terrestrial Climatology, Journal of Climate, 12, 829–856, https://doi.org/10.1175/15200442(1999)012<0829:RTCSTC>2.0.CO;2, 1999.

New, M., Hulme, M., and Jones, P.: Representing Twentieth-Century Space–Time Climate Variability. Part II: Development of 1901–96 Monthly Grids of Terrestrial Surface Climate, Journal of Climate, 13, 22172238, https://doi.org/10.1175/15200442(2000)013<2217:RTCSTC>2.0.CO;2, 2000.